# BrainBits: How Much of the Brain are Generative Reconstruction Methods Using?

**David Mayo[1*]**     **Christopher Wang[1*]**     **Asa Harbin[2*]**     **Abdulrahman Alabdulkareem[1]**

**Albert Shaw[3]**          **Boris Katz[1]**          **Andrei Barbu[1]**

[1]MIT CSAIL, CBMM          [2]MIT Lincoln Laboratory          [3]Google DeepMind

## Abstract

When evaluating stimuli reconstruction results it is tempting to assume that higher fidelity text and image generation is due to an improved understanding of the brain or more powerful signal extraction from neural recordings. However, in practice, new reconstruction methods could improve performance for at least three other reasons: learning more about the distribution of stimuli, becoming better at reconstructing text or images in general, or exploiting weaknesses in current image and/or text evaluation metrics. Here we disentangle how much of the reconstruction is due to these other factors vs. productively using the neural recordings. We introduce BrainBits, a method that uses a bottleneck to quantify the amount of signal extracted from neural recordings that is actually necessary to reproduce a method's reconstruction fidelity. We find that it takes surprisingly little information from the brain to produce reconstructions with high fidelity. In these cases, it is clear that the priors of the methods' generative models are so powerful that the outputs they produce extrapolate far beyond the neural signal they decode. Given that reconstructing stimuli can be improved independently by either improving signal extraction from the brain or by building more powerful generative models, improving the latter may fool us into thinking we are improving the former. We propose that methods should report a method-specific random baseline, a reconstruction ceiling, and a curve of performance as a function of bottleneck size, with the ultimate goal of using more of the neural recordings.

## 1 Introduction

Applying powerful generative models to decoding images and text from the brain has become an active area of research with many proposed methods of mapping brain responses to model inputs. A race between publications is driving down the reconstruction error to produce higher fidelity images and text [17, 27, 28]. It could be easy to assume that as the field gets better at reconstructing stimuli, we will simultaneously be getting better at modeling vision and language processing in the brain. We argue that this is not necessarily the case.

There are several reasons why a method might have higher quality reconstructions yet actually require the same or less signal from the brain. For example, a much larger model can learn a stronger prior over the space of images and text, so even if it were given less information from the brain, it might produce better reconstructions. In particular, a generative model might become more fine-tuned toward the distribution of images and text that are used in standard datasets. This is problematic because so few open neuroscience datasets exist, and even fewer at the scale that would

---

*Equal contribution. Code available at https://github.com/czlwang/BrainBits. Correspondence to DM and CW at {dmayo2, czw}@mit.edu and AH at asaharbin@ll.mit.edu.

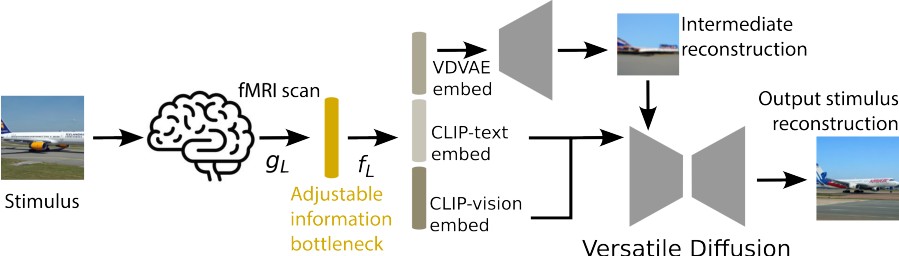

Figure 1: **BrainBits bottlenecking framework as applied to BrainDiffuser.** The goal of image reconstruction is to generate an image based on brain signal. The brain signal is mapped to a hidden vector (gold) by a compression mapping $g_L$, which is then used to predict VDVAE, CLIP-text, and CLIP-vision latents via a mapping $f_L$. As in [18], these latents are used to produce the final reconstruction. In our studies, we restrict the information available from the brain by varying the dimension of the hidden vector.

enable this research. It is easy to inadvertently overfit a model and over-optimize for the particular biases of the standard benchmarks, never mind explicitly tuning the parameters. Finally, there is the separate, confounding issue of how to best evaluate the reconstructions. Even the best intentioned modeling approaches on novel data can run afoul of the extremely limited image and text evaluation methods that we have today. Later in the manuscript, we demonstrate the importance of appropriately calibrating and understanding the shortcomings of these methods.

Given that better decoding need not explain more of the brain, we create the first metric to measure this: BrainBits. BrainBits measures how reconstruction performance varies as a function of an information bottleneck. We learn linear mappings from the neural recordings to a smaller-dimensional space, optimizing the reconstruction objective of each method.

The result of running BrainBits on state-of-the-art reconstruction methods is striking: a bottleneck that is a small percentage of the full brain data size is sufficient to guide the generative models towards images of seemingly high fidelity. For fMRI, the entire brain volume often has on the order of 100K total voxels and about 14K voxels in the visual area, which is what the methods we report here use. We find that a reduction through a bottleneck of only 30 to 50 dimensions provides the vast majority of the performance of a reconstruction method depending on the metric.

BrainBits enables us to disentangle the contributions of the generative model's prior and the signal extracted from neural recordings when evaluating models. This is critical to soundly using stimuli reconstruction as a tool for making neuroscientific progress. We would like reconstruction methods that explain more of the brain rather than merely relying on better priors. In particular, we propose three components: to produce a method-specific random baseline that uses no neural recordings, to compute a method-specific reconstruction ceiling, and to compute reconstruction performance as a function of the bottleneck size.

Ideally, models would achieve near full performance only with large bottlenecks, showing that they are relying on the neural signal for their performance. Models that have high random baselines and then exploit only a few bits of information from the brain in order to achieve their, at first glance, impressive performance, are doing so largely from their prior. And while all the examples of BrainBits we provide here use fMRI, the method can be applied to any neural recording modality. BrainBits also provides interpretability, by showing which brain areas contribute to decoding as a function of the bottleneck size and making the activity of these regions available for probing via decoder.

Our contributions are:

1. BrainBits, a novel method that uncovers that models use very little of the neural signal to achieve their performance.
2. An application of BrainBits to three recent stimulus reconstruction methods, two for vision and one for language decoding.
3. An investigation of which brain areas are most relied upon for reconstruction, made possible by our interpretable linear bottleneck design
4. An analysis of features which are available in the bottlenecks, how quickly those features saturate, and which features can contribute to performance when using more of the brain recordings.

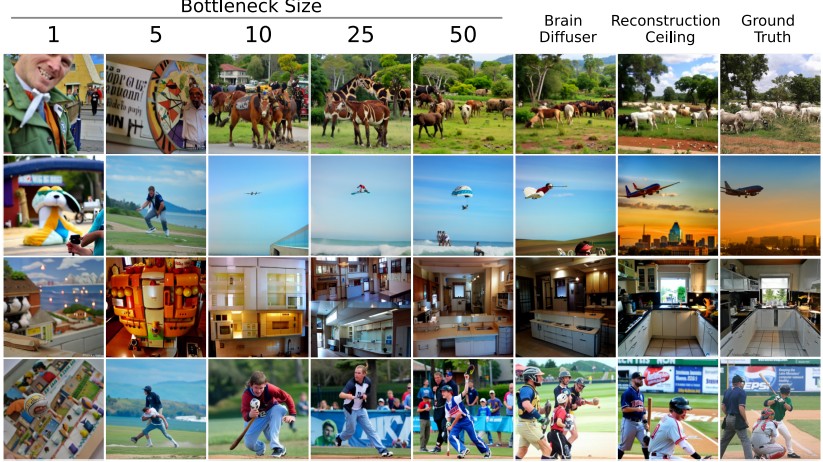

(a) BrainDiffuser

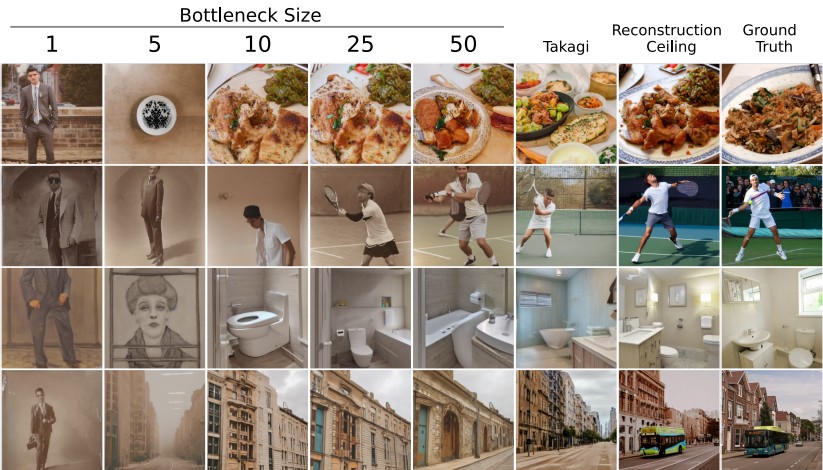

(b) Takagi & Nishimoto 2023

| Bottleneck | Text output |
| --- | --- |
| 100 | *we both lived in different countries she moved out of state to college to study abroad in the us we never dated for more than a year and we only...* |
| 1000 | *same city i had just graduated from a university in a state i lived in on a scholarship that was much higher than what most schools have and the system...* |
| Full | *the same university we had just moved from the us to a different state she had a degree in psychology and a job that i was not one for but...* |
| Ground truth | *we're both from up north we're both kind of newish to the neighborhood this is in florida we both went to college not great colleges but man we graduated and...* |

(c) Tang et al. 2023

Figure 2: **High quality stimuli can be reconstructed from a fraction of the data.** Shown here are images and text reconstructed for several bottleneck sizes using our BrainBits approach. Images and text are shown for subject 1 for all three methods. Examples where the original methods could reasonably reconstruct the stimuli were chosen; the same images for both visual methods are shown in the appendix. As the bottleneck dimension increases, the accuracy of the reconstruction increases. Although there are differences between the full and bottlenecked ($d = 50$) results, the reconstructions are surprisingly comparable, despite the fact that the full reconstruction methods have $> 14,000$ voxels available to them. Text reconstructions are harder to evaluate in this qualitative manner, later we present a quantitative evaluation.

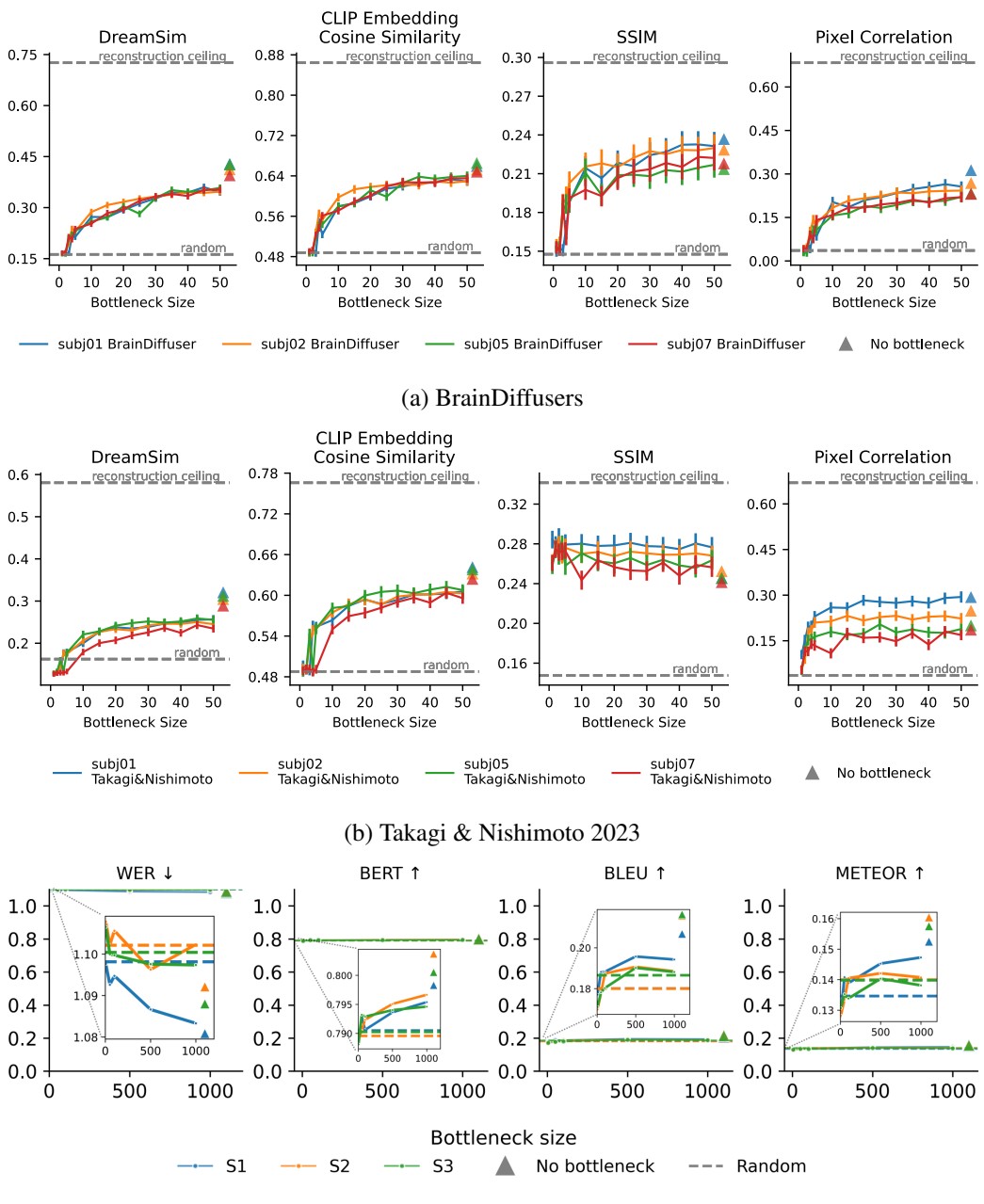

(a) BrainDiffusers

(b) Takagi & Nishimoto 2023

(c) Tang et al. 2023 with insets zooming in for clarity

Figure 3: **Quantifying the fraction of the data needed to reconstruct stimuli.** While different metrics present slightly different pictures of model performance, most performance is reached by about 20 32-bit floating point numbers and essentially all performance is reached by about 50. Vision reconstruction methods (a,b) are significantly better than language reconstruction methods (c). And although language methods appear to use very large bottlenecks, as a fraction of the data available, they are comparable (vision methods presented here use only voxels in the visual cortex). Relative to vision, language methods are much closer to the random baseline and have a longer way to go, as shown by the inset. This also reveals a limitation of the resolution of the BrainBits approach: there is not much room for bottlenecking when performance is near the random baseline and metrics lack a well calibrated scale. Across different metrics, both low level metrics (like pixel correlation and word error rate) and high level metrics (like DreamSim and BERT) the message is the same: models asymptote quickly.

## 2 Related work

Several recent works have focused on predicting the latent features of deep, pretrained, generative models from fMRI data in order to reconstruct corresponding stimuli. Han et al. [9] introduced a technique that projects fMRI recordings to the bottleneck layer of an image-pretrained variational autoencoder [12] and reconstructs the stimulus via the decoder network. Similarly, [24] learns projections to the input space of the generator network belonging to a pretrained generative adversarial network [8]. Given the success of models such as Dall-E [22], more recent work has focused on learning mappings to the latent space of large diffusion models [2, 23, 25, 26, 30]. Furthermore, approaches such as [5] and [16] leverage recent generative models for the multimodal case, decoding both images and captions.

This family of methods has been facilitated by the growth of fMRI datasets containing pairs of stimuli and recorded neural data, the current largest of which is the publicly available Natural Scenes Dataset (NSD) [1]. The Natural Scenes Dataset contains fMRI recordings of multiple subjects cumulatively viewing tens of thousands of samples from the Microsoft CoCo dataset [14]. Given its increased size relative to previous similar datasets [11], it presents more potential for data-driven neural decoding. As a result, it is a popular choice for many recent methods [6, 29], and we select it for our analysis.

Numerous metrics for measuring reconstruction fidelity have been proposed. In the visual domain these include pixel correlation, SSIM, CLIP similarity, and DreamSim [7]. For language these include word error rate (WER), BLEU, METEOR, and BERTScore [32]. None take into account the prior knowledge that modern models have built into them.

## 3 Approach

Given a reconstruction method $f$ that maps brain data $X$ to images $Y$, we seek to determine how much the quality of the images $\hat{Y} = f(X)$ depends on the brain signal. We do this by placing restrictions on information flow, and then examining the resulting reconstructions. This restriction is operationalized by a bottleneck mapping $g_L$ that compresses the brain data to a vector of smaller dimension. Specifically, let $Y = \{y_i\}$ where $y_i$ is an individual original image corresponding to the brain data response $x_i$. Then, as stated, our aim is to find the best reconstruction achievable for a given restriction $L$, where reconstruction quality is scored by some metric $s(\cdot, \cdot)$:

$$\max_{g_L} \sum_i \left| s(f(g_L(x_i)), y_i) \right| \tag{1}$$

This allows us to produce a curve of reconstruction quality as a function of $L$. We restrict our attention to linear transformations, $g_L$, to find the interpretable mappings that yield the best reconstruction quality.

Model performance lies in a range between model-specific randomly generated images and a model-specific ceiling. To compute a model's random performance, we run bottleneck training and reconstruction, substituting the original brain data with synthetic data generated according to $\mathcal{N}(0, 1)$. The purpose of this baseline is to obtain a set of images that reflect the generative prior of the model without any input from the brain.

To compute a ceiling on the image reconstructions, BrainDiffuser and Takagi et al 2023, we run the complete original reconstruction pipeline, but substitute the ground truth image latents instead of using the latents as predicted from the brain data. We do this to obtain the reconstructions as produced by the generative models, had the target latents been predicted perfectly. No analogous ceiling procedure exists for the language reconstruction method, Tang et al 2023, which is based on scoring word predictions via an encoder model.

## 4 Experiments

We adapt three state-of-the-art methods BrainDiffusers [17], Takagi & Nishimoto [27], and Tang et al. [28] to compute BrainBits; the first two are vision reconstruction methods and the last is a language reconstruction approach. In each case, BrainBits is computed in the same way, but it is computed jointly with the optimization for each method. This means that BrainBits is not a simple pre- or post-processing step or function call, it must be integrated into the method, which at times can

require updates to the optimizer being used, effectively calling for a port from standard regression libraries to a deep learning framework. We optimize reconstruction for varying bottleneck sizes, and evaluate the resulting reconstructions on the standard metrics, including the ones used by the authors, as well as a new metric that has since been proposed, DreamSim, in order to show that BrainBits produces the same message regardless of the metric chosen or modality of the reconstruction. Below, we describe each method and how BrainBits was computed.

### 4.1 BrainDiffusers

The original BrainDiffusers uses fMRI data from the Natural Scenes Dataset [1] (see Section 2), in which the brain volumes have been masked specifically to only include the visual areas ($\mu = 13930$ voxels). In the BrainDiffusers approach, regressions are fitted to map the fMRI data to latent representations of the corresponding images, namely VDVAE [3] and CLIP [21] embeddings of the images. An additional regression is fitted to predict CLIP embeddings of the corresponding COCO captions. The predicted VDVAE latent is used to produce a coarse version of the image. Then, the predicted VDVAE image, the predicted CLIP-text embedding, and the predicted CLIP-vision embeddings are given as input to versatile-diffusion [31], which produces the final predicted image. Complete details are given in the original paper [17].

Our approach to bottlenecking BrainDiffusers is shown in Figure 1. For a given bottleneck size $L$, we learn a mapping $g_L$ from the fMRI input to a $L$-dimensional vector. From this vector, we learn a mapping to the image and text embedding targets. See (Appendix A.1: Training Bottlenecks) for training details.

### 4.2 Takagi & Nishimoto

This approach is broadly similar to BrainDiffuser in that the same dataset is used, and the same approach of mapping fMRI signal to embedding targets is used. However, there are a few key differences. First, separate mappings are learned for different parts of the brain. The early visual area is mapped to the latent representation space of a VAE, and the outputs of this mapping are passed through the VAE's decoder to produce a course reconstruction of the image stimuli.

Another mapping is fit from a concatenation of the early, ventral, midventral, midlateral, lateral, and parietal regions to BLIP [13] embeddings of the image stimuli. Predictions of these embeddings are decoded into text that is then used along with the coarse image reconstruction to guide image generation with Stable Diffusion [19]. Since two separate mappings are learned, we insert two bottlenecks trained separately and keep their size the same.

### 4.3 Tang et al.

We show how our benchmark framework can be extended to other modalities by also making a study of an fMRI-to-language reconstruction approach. In the original approach, an encoding model is fit to map GPT [20] embeddings to brain activity. Then, at inference time, the decoder takes the brain activity as input and uses GPT to auto-regressively propose candidate predictions for the next word. The word with the highest likelihood, as computed by the encoding model, is accepted as the next word in the sequence. We follow the original method in separating the training and decoding steps. We insert the information bottleneck by first learning a mapping from brain activity to a compressed vector that is mapped to a GPT text embedding. Instead of using the raw brain activity as input, we use the resulting bottleneck representations, and run the rest of the pipeline without modification. This method has the fewest changes and simplest adaptation to BrainBits. Tang et al. reconstruct language from perceived speech, imagined speech, perceived movie, and perceived multi-speaker speech (see [28]), and we report performance averaged across these tasks. Additional details are given in the appendix.

## 5 Results

We use standard metrics for image [18] and text [28] construction. When reporting CLIP score, we compute the absolute cosine similarity between images rather than an average image rank sorted by CLIP similarity; this makes results comparable to CLIPScore [10] modulo a constant scaling factor.

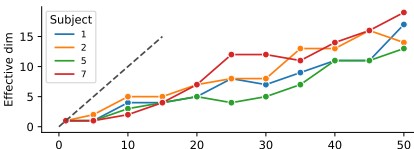
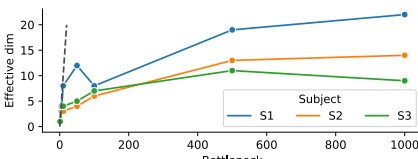

Figure 4: **How large are the bottlenecks?** Even though the bottleneck representations have $L$ dimensions, it is not necessarily the case that all dimensions are used by the bottleneck mapping. For both language and vision, we can measure the effective dimensionality to get a sense for how much of the channel capacity is being used. For BrainDiffuser, the effective dimensionality is comparable to the bottleneck size, showing that information is being extracted from the neural recordings up to about 15-20 dimensions For language bottlenecks, effective dimensionality remains low showing that little of the channel capacity, and therefore little of the neural signal, is being used.

We also consider similarity as measured by DreamSim [7]. Including additional metrics demonstrates that the BrainBits message is independent of the metric used: models use relatively little of the neural recordings to achieve the vast majority of their performance.

As described above, we insert information bottlenecks into two vision and one language reconstruction method and vary the bottleneck size while optimizing on the original objectives. Having learned these bottleneck mappings, we then investigate the resulting reconstructions. We can also study the representations learned by the bottleneck mappings themselves, and we compute the effective dimensionality of the bottleneck representations, the types of information decodable from the representations, and the weight that each bottleneck mapping places on different regions of the brain.

**How much information is needed to reconstruct an image or text?** Qualitative results are shown in Figure 2 while quantitative results are shown in Figure 3. For BrainDiffuser a bottleneck of size 50 achieves 75%, 95%, 100%, and 89% of the original performance as measured by DreamSim, CLIP cosine similarity, SSIM, and pixel correlation. This is a reduction of a factor of approximately 300, given that the method starts with approximately 14,000 voxels, to achieve most to all of the reconstruction performance. A similar trend holds for the Takagi & Nishimoto method.

Chance performance is high for both visual reconstruction methods because the models learn strong priors over the data. More surprising is the relatively low ceiling that both methods have. Even inserting the best possible latents, methods are very limited in their ability to maximize these metrics.

For language reconstruction, the original uses whole-brain fMRI; approximately 90,000 floating point numbers. A bottleneck of size 1000 is sufficient to recover 50% of the performance, averaged across subjects, as measured by BERT. This bottleneck achieves 26% and 20% of the original performance, as measured by BLEU and METEOR respectively, and the WER is comparable. Chance performance is similarly surprisingly good, with a WER of approximately 1.1, BLEU of approximately 0.18, METOR of approximately 0.14, and BERT score of approximately 0.79; these vary slightly by subject. As discussed in the conclusion, a limitation of BrainBits is that the bottleneck size may be exaggerated in cases like these where performance is relatively close to chance.

**How effectively are the bottlenecks used?**

The bottleneck dimension is an upper bound on the information being extracted from the brain. A 50-dimensional bottleneck may contain a much lower dimensional signal; see Figure 4. We use the number of principal components needed to explain at least $95\%$ of the variance, as a measure of the effective dimensionality of the bottlenecked representations [4]. For BrainDiffuser, a bottleneck of size 50 has an effective dimension of about 16, averaged across subjects. For comparison, the average effective dim of the fMRI inputs is $2,257$ (see supplementary Figure 16). Finally, language models have a much smaller effective dimension: a bottleneck of size 1,000 has roughly an effective dimensionality of 5 to 20 depending on the subject.

**What regions of the brain matter most?**

We plot the weights of the bottleneck mapping back onto the brain for BrainDiffuser; see Figure 5. The vast majority of the weight is assigned to voxels in the periphery of the early visual cortex. As the model has access to larger bottlenecks, it continues to assign more importance to these areas rather than including new areas. One would ideally like models to expand the brain areas that they use effectively as the bottleneck size increases.

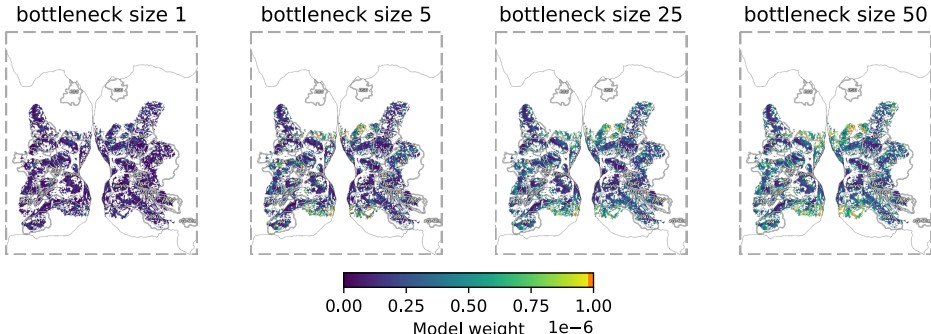

Figure 5: **What areas of the brain help reconstruction the most?** Models quickly zoom in on useful areas even at low bottleneck sizes. Note that for clarity the color bar cuts off at 1e-6, values above that are all orange. In this case BrainDiffuser on subject 1 attends to peripheral areas of the early visual system. As the bottleneck size goes up models exploit those original areas but do not meaningfully expand to new areas. Ideally, one would hope to see more of the brain playing an important role with larger bottleneck sizes; this is not what BrainBits uncovers.

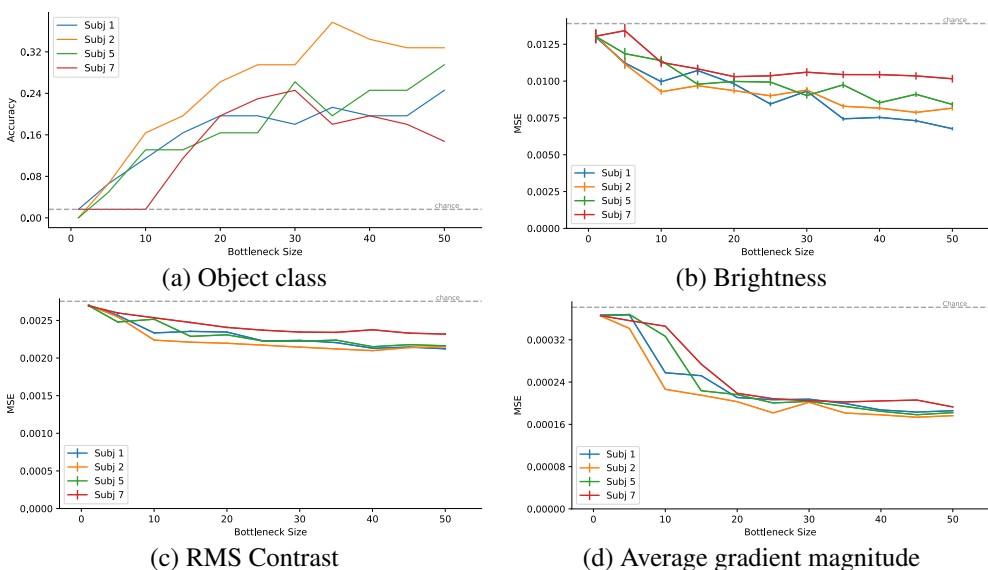

Figure 6: **What information do bottlenecks contain?** For the BrainDiffusers approach we compute the decodability of four different features (object class, brightness, RMS contrast, and the average gradient magnitude) as a function of bottleneck size. Object class refers to decoding the class of the largest object in the image; often the focus of the image. The average gradient magnitude is a proxy for the edge energy in the image. Dashed lines in plot (a) indicate 1-out-of-61 classification chance, 1.6%. Dashed lines on plots (b, c, d) indicate the metric's MSE distance from the average metric value on the training set. Larger bottlenecks are needed to extract more object class information above chance. Edge energy, brightness and contrast are mostly exhausted early. Looking at features as a function of bottleneck size can reveal what types of interpretable features models learn, offering some explanation as to why performance goes up as a function of bottleneck size.

**What do the bottlenecks contain?**

We attempt to decode visual features from bottlenecks of different sizes for BrainDiffuser; see Figure 6. Lower-level features like contrast, brightness, and edge energy are available very quickly at low bottleneck sizes and are also largely exhausted fairly quickly. High-level features like object class may be driving performance as the bottleneck size increases.

# 6 Limitations

BrainBits has a number of limitations. It requires rerunning the decoding process several times. This can be expensive depending on the method. It is also not plug-and-play, since it must be optimized jointly with the reconstruction method. While a simple fixed compression scheme such as PCA can be used for a rough estimate, jointly optimizing the bottleneck and the reconstruction method can be a more difficult optimization problem requiring some manual attention. Depending on the method, BrainBits may be inserted at different points, for example, a method may have several steps that extract information from the brain. All of this prevents BrainBits from being a simple library call. Models must be adapted to compute BrainBits instead, although this adaptation is generally simple.

The resolution of BrainBits depends in part on sweeping the bottleneck size, but it also depends on precisely how that bottleneck is computed. We only consider a linear bottleneck to avoid adding meaningful computations to the model. One could also consider methods that employ vector quantization, which we intend to do in the future. The linear approach we take here has difficulty training with bottleneck sizes that are smaller than one float. A vector quantization method would likely work better for such small bottlenecks. Although, small bottlenecks are perhaps not that interesting given that the goal is to explain more of the brain.

In general, current image and text metrics are limited and make understanding reconstruction performance difficult. This situation is made worse by the fact that random performance can be high when models have strong priors. And even more so when those priors allow models to perform well with little added information from the brain. Computing these quantities is critical for understanding where we are in terms of absolute performance and explaining representations in the brain.

**Computational requirement and code availability** All mappings and image generations were computed on two Nvidia Titan RTXs over the course of a week. Code is available at https://github.com/czlwang/BrainBits.

# 7 Discussion and Conclusion

The fidelity of a reconstruction depends on the priors of the generative model being used and the amount of useful information extracted from the brain. Through visual inspection or from reconstruction metrics, one can easily be fooled into thinking that because the results appear high fidelity, they must leverage large amounts of brain signal to recover such details. BrainBits reveals otherwise. Relatively little of the neural recordings are used, and many of the produced details can be attributed to the generative prior of the diffusion model. To get a clearer understanding of how to evaluate these reconstructions, we propose a realistic random baseline based on the generative prior, as well as a reconstruction ceiling based on what it is possible to decode with perfect latent prediction. The random baseline achieved by generative models is far higher that most would expect and the reconstruction ceiling on some metrics is far lower than expected.

The priors create a much narrower range of performance than one might expect. For example, BrainDiffusers has an effective range of approximately 0.15 to 0.75 DreamSim, 0.48 to 0.88 CLIP, 0.15 to 0.3 SSIM, and 0.05 to 0.7 Pixel Correlation. These ranges depend entirely on the models employed, and are a reflection of the priors of the models. Reporting at least the floor is important for contextualizing results; this is not reported in most prior work.

Bottlenecks appear to exploit information in order, from low-level features, brightness and contrast, to mid-level features, edge energy, to high level features, object class. Vision models focus on the same region of the brain regardless of bottleneck: early visual cortex. Higher-level features should become more disentangled and easier to take advantage of in later parts of the visual system. This does not appear to be useful to current models.

Goodhart's law states: "When a measure becomes a target, it ceases to be a good measure". With the advent of high resolution reconstructed image stimuli, we as a field may be tricked into believing that we have become better at understanding visual processing in the brain. We may be tempted into further optimizing the quality of the reconstructed images in this service. But this is an inappropriate target if neuroscientific insight is our goal. We emphasize the importance of a BrainBits analysis for all neuroscientific studies of stimulus reconstruction to quantify the true contribution of the brain to reconstructions.

## 7.1 Acknowledgements

We would like to thank Colin Conwell and Brian Cheung for their helpful feedback and discussion about our experiments.

This work was supported by the Center for Brains, Minds, and Machines, NSF STC award CCF-1231216, the NSF award 2124052, the MIT CSAIL Machine Learning Applications Initiative, the MIT-IBM Watson AI Lab, the CBMM-Siemens Graduate Fellowship, the DARPA Artificial Social Intelligence for Successful Teams (ASIST) program, the DARPA Mathematics for the DIscovery of ALgorithms and Architectures (DIAL) program, the DARPA Knowledge Management at Scale and Speed (KMASS) program, the DARPA Machine Common Sense (MCS) program, the United States Air Force Research Laboratory and the Department of the Air Force Artificial Intelligence Accelerator under Cooperative Agreement Number FA8750-19-2-1000, the Air Force Office of Scientific Research (AFOSR) under award number FA9550-21-1-0399, the Office of Naval Research under award number N00014-20-1-2589 and award number N00014-20-1-2643, and this material is based upon work supported by the National Science Foundation Graduate Research Fellowship Program under Grant No. 2141064. This material is based on work supported by The Defense Advanced Research Projects Agency under Air Force Contract No. FA8721-05-C-0002 and/or FA8702-15-D-0001. Any opinions, findings, and conclusions or recommendations expressed in this material are those of the author(s) and do not necessarily reflect the views of The Defense Advanced Research Projects Agency. The views and conclusions contained in this document are those of the authors and should not be interpreted as representing the official policies, either expressed or implied, of the Department of the Air Force or the U.S. Government. The U.S. Government is authorized to reproduce and distribute reprints for Government purposes notwithstanding any copyright notation herein.

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

# A   Appendix

## A.1   Training bottlenecks

**Case study: BrainDiffuers**   The original BrainDiffusers method learns separate mappings to the VDVAE, CLIP-text and CLIP-vision latents. We predict all embeddings simultaneously. We use an MSE objective, and weight the loss on the predicted VDVAE, CLIP-text, and CLIP-vision targets with $[1, 2, 4]$ respectively. We train our network with a batch size $b = 128$, an AdamW optimizer [15], a weight decay of $wd = 0.1$ and a learning rate of $lr = 0.01$. We train for 100 epochs and use the weights with the best validation loss at test time.

**Case study: Tang et al. 2023**   Tang et al. use the fMRI data to both predict the timing and the content of language. Because we are interested in the semantic decoding, we use the original models of timing, and use the bottleneck representations to predict the words itself. We train our bottleneck with a batch size of $b = 512$, learning rate $lr = 5e - 4$, and the AdamW optimizer. We train for 100 epochs and use the weights with the best validation loss at test time. For evaluation, we report the average of scores across windows, (see [28] for details).

**Takagi et al 2023**   Takagi et al. learn mappings from the early visual area to VAE latents of the stimuli [27]. They also learn mappings from the early, ventral, midventral, midlateral, laterial, and parietal regions to the latents of a BLIP image encoder [13]. We learn a separate bottleneck for each mapping with the MSE objective. We use the AdamW optimizer [15] and perform hyperparameter search over learning rates in $[1e - 5, 1e - 4, 1e - 3, 1e - 2]$ and weight decays in $[1e - 3, 1e - 2, 1e - 1, 5e - 1]$ .

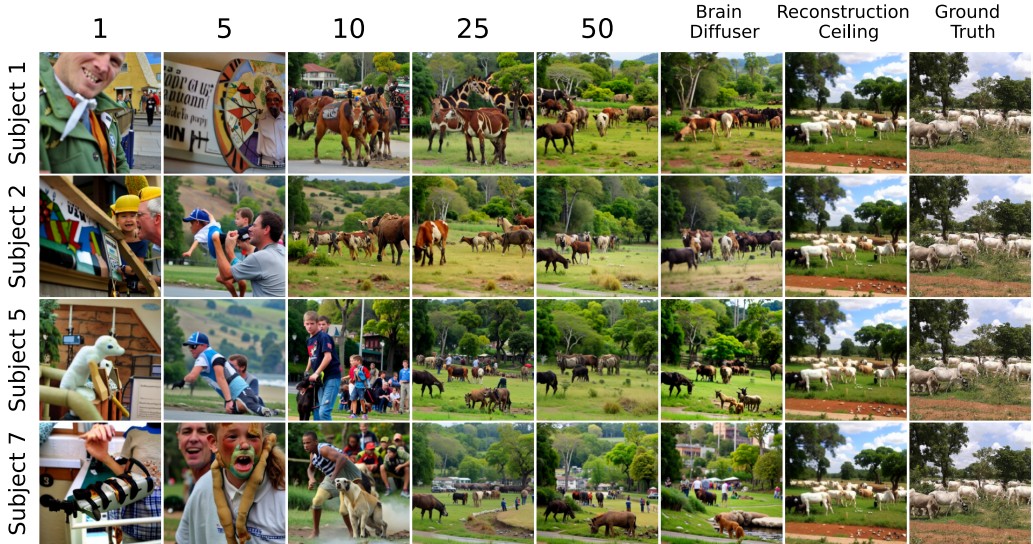

Figure 7: Samples for all subjects in BrainDiffuser, for a single image

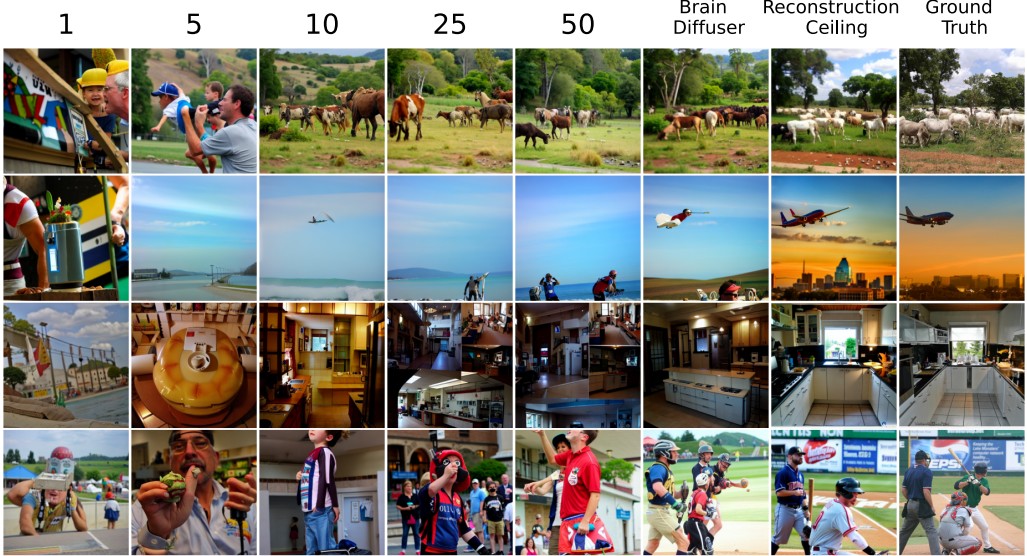

Figure 8: The same images shown for subject 1 in Figure 2a are shown here for subject 2

| 1 | 5 | 10 | 25 | 50 | Brain Diffuser | Reconstruction Ceiling | Ground Truth |

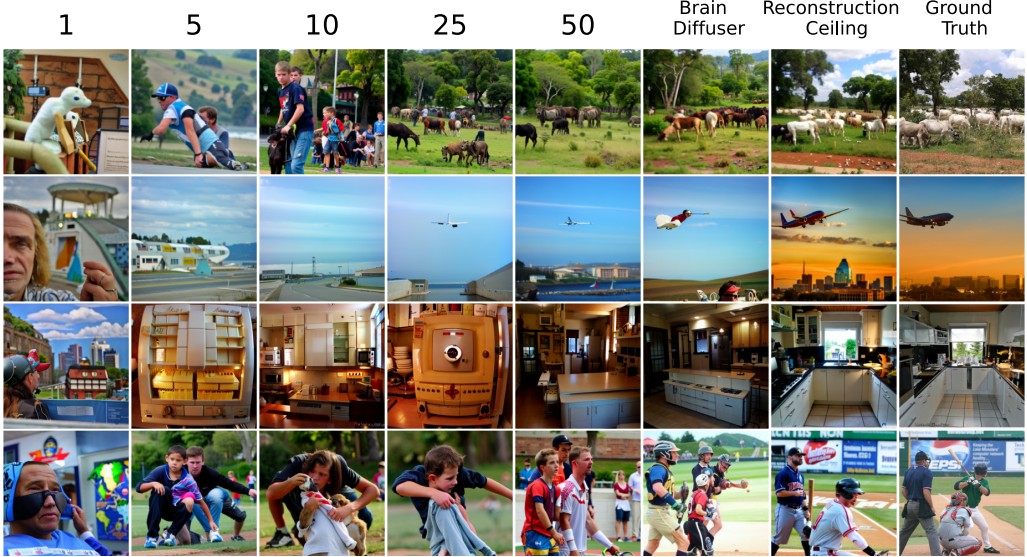

Figure 9: The same images shown for subject 1 in Figure 2a are shown here for subject 5

| 1 | 5 | 10 | 25 | 50 | Brain Diffuser | Reconstruction Ceiling | Ground Truth |

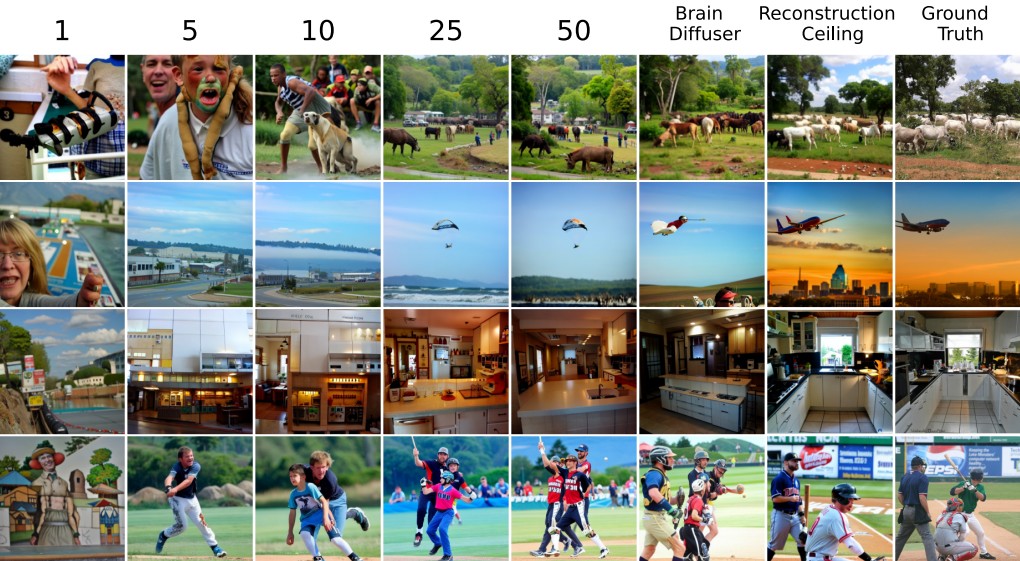

Figure 10: The same images shown for subject 1 in Figure 2a are shown here for subject 7

## A.3 Projecting bottlenecks onto the brain

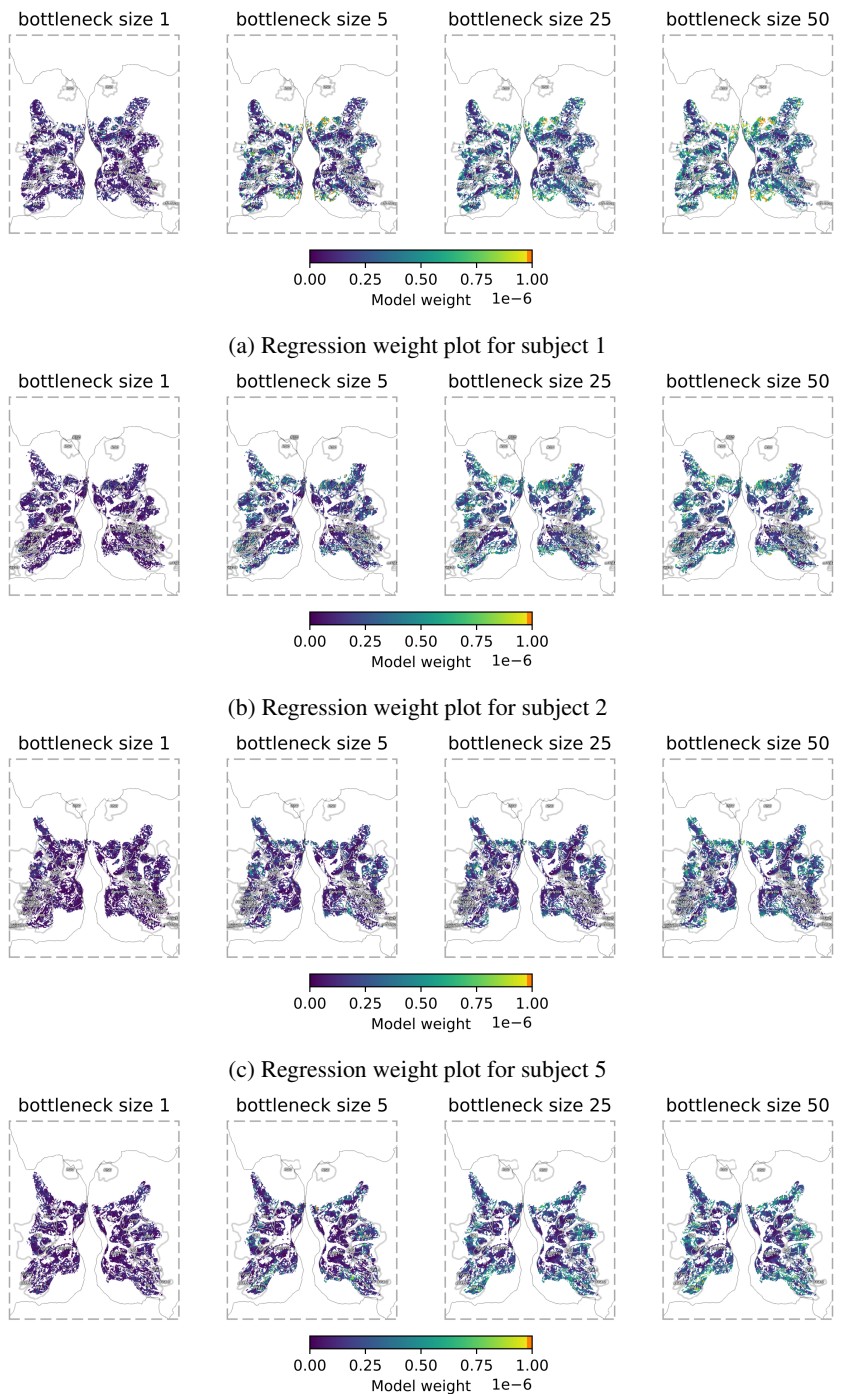

(a) Regression weight plot for subject 1

(b) Regression weight plot for subject 2

(c) Regression weight plot for subject 5

(d) Regression weight plot for subject 7

Figure 11: Regression weight plots for each subject

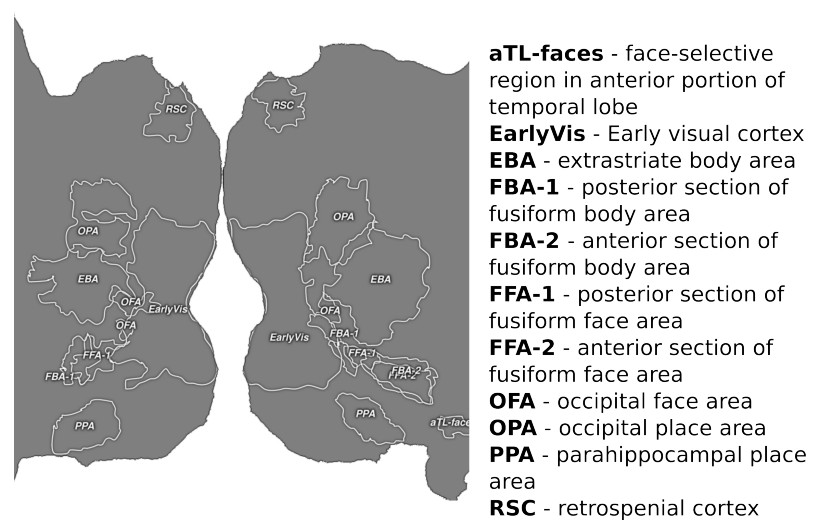

**aTL-faces** - face-selective region in anterior portion of temporal lobe
**EarlyVis** - Early visual cortex
**EBA** - extrastriate body area
**FBA-1** - posterior section of fusiform body area
**FBA-2** - anterior section of fusiform body area
**FFA-1** - posterior section of fusiform face area
**FFA-2** - anterior section of fusiform face area
**OFA** - occipital face area
**OPA** - occipital place area
**PPA** - parahippocampal place area
**RSC** - retrospenial cortex

Figure 12: Legend of ROIs for main text salience maps in Figure 5 and Figure 11.

## A.4 Text reconstruction examples

| Bottleneck | Text output |
| --- | --- |
| 1 | *didn't want to be alone i had to get out of there i ran to the bathroom and threw up i was so ashamed i didn't even know what to do i just sat there for a few minutes and then i went to the bathroom and i was crying i had never been* |
| 5 | *this one incident that will never be forgotten and it's only ever been reported in the us and i'm sure it's been told to me and probably all of my family i have never been asked for my name i don't have one i am only allowed to show my id i have no* |
| 50 | *high school after the summer after graduating from a university with no major in life we were both a bit socially awkward but it was definitely one of the best things ever and we both loved each other for that alone we never got married or anything so that was a first and i* |
| 100 | *we both lived in different countries she moved out of state to college to study abroad in the us we never dated for more than a year and we only talked on the phone and i don't remember her name i know her from my online profile which is pretty cool because i can* |
| 500 | *went to college in a town about ten hours away from campus and lived there for years and never met anyone who was good looking or interesting but i think there were a few that were i got the idea to date someone with a degree in math and i had to learn that* |
| 1000 | *same city i had just graduated from a university in a state i lived in on a scholarship that was much higher than what most schools have and the system was very strict so my mom basically forced me to pay for a college degree in math and i couldn't afford the same in* |
| Full | *the same university we had just moved from the us to a different state she had a degree in psychology and a job that i was not one for but it seemed she was the kind of girl that got into college for her degree in computer science i had to learn the language* |
| Ground truth | *we're both from up north we're both kind of newish to the neighborhood this is in florida we both went to college not great colleges but man we graduated and i'm actually finding myself a little jealous of her because she has this really cool job washing dogs she had horses back home and she really* |

Table 1: Expanded full text for the subject 1 samples shown in Figure 2c

| Bottleneck | Text output |
|---|---|
| 1 | *worst if she doesn't want to have sex with you and you end up getting hurt or worse if you don't have a girlfriend and she decides to go out with another guy then you have to be the one to tell her that you are not interested in her and that she should just go back to* |
| 5 | *and now i'm not so sure i should have just given up the whole idea of living out of a truck in a trailer my first thought is to buy a house and move out the second is that it's not even that big a deal we live in the same state as you i can't understand why* |
| 50 | *phd program at a small college in a suburb of my hometown in the city i have lived in for years and am the youngest female graduate there i've never been married but have done a few times so i was wondering if you have ever had a girl come on to you so hard and it felt* |
| 100 | *same city i had just gotten married to a girl who has recently graduated from a university in a state she has no money for a lawyer who will never be able to afford one i also know this because my sister has an aunt who has a degree in accounting that has the ability to put up* |
| 500 | *up in a small rural town in the west coast of the state i lived in was not an alcoholic nor a drug addict i didn't smoke weed i was never a bad person but i had a problem with drinking for example my father drank more than a bottle of wine because he didn't like his body* |
| 1000 | *high school was in a small town near where i grew up in my hometown was the biggest city in the state i knew most people were good looking and all that but i had no idea the average guy was this skinny nerd with no social skills or a brain and he was afraid of heights because* |
| Full | *grew up in the south and the area i live in was a major city in my country we went to college at the same university i am a pretty smart kid but i was never the kind of guy to actually work at a computer science lab or study programming and i think he should be a* |
| Ground truth | *we're both from up north we're both kind of newish to the neighborhood this is in florida we both went to college not great colleges but man we graduated and i'm actually finding myself a little jealous of her because she has this really cool job washing dogs she had horses back home and she really* |

Table 2: Expanded full text for the same stimulus as shown in Figure 2c and Table 2, but for subject 2

| Bottleneck | Text output |
|---|---|
| 1 | *name but she didn't answer i went to the bathroom and i was crying i had no idea what to do i didn't know what i could do to help her i just wanted to get her out of there i was so scared i couldn't even look at her i just* |
| 5 | *damn movie theater and watch my wife die right there and i feel so horrible but i'm just so angry and so incredibly sad i hate her for not telling me that i am the best and the most beautiful and that she should never have to work so hard and so* |
| 50 | *the manager talking to a guy i wasn't familiar with on the phone we made it to the bar and there were at least of us and i remember some of the conversations we had and the whole place seemed to be filled with the same stuff the only difference was that* |
| 100 | *in a large city with about hundred students from a variety of major and regional universities all graduating with degrees in physics history and math in engineering and biology all of this was based around a very big and complicated system of programming that had been built by very very small and* |
| 500 | *a university which was in a suburb of a city with a small town of about million students all graduating with degrees in physics i was one of the youngest of them all of a sudden a very powerful and intelligent individual with a phd in physics from a small liberal arts* |
| 1000 | *in a city with about thousand students all graduating from a prestigious college in the middle east there are hundreds of universities all over the country where i was raised from what i saw was a very large area with lots of very liberal arts majors and an average high school level* |
| Full | *he was born in a city that was mostly rural but was in the south of the us and was an engineer and a teacher all of which i had a good relationship with she was also very interested in working in a computer science program but had a job in another* |
| Ground truth | *we're both from up north we're both kind of newish to the neighborhood this is in florida we both went to college not great colleges but man we graduated and i'm actually finding myself a little jealous of her because she has this really cool job washing dogs she had horses back home and she really* |

Table 3: Expanded full text for the same stimulus as shown in Figure 2c and Table 3, but for subject 3

## A.5 Two-Way Evaluation

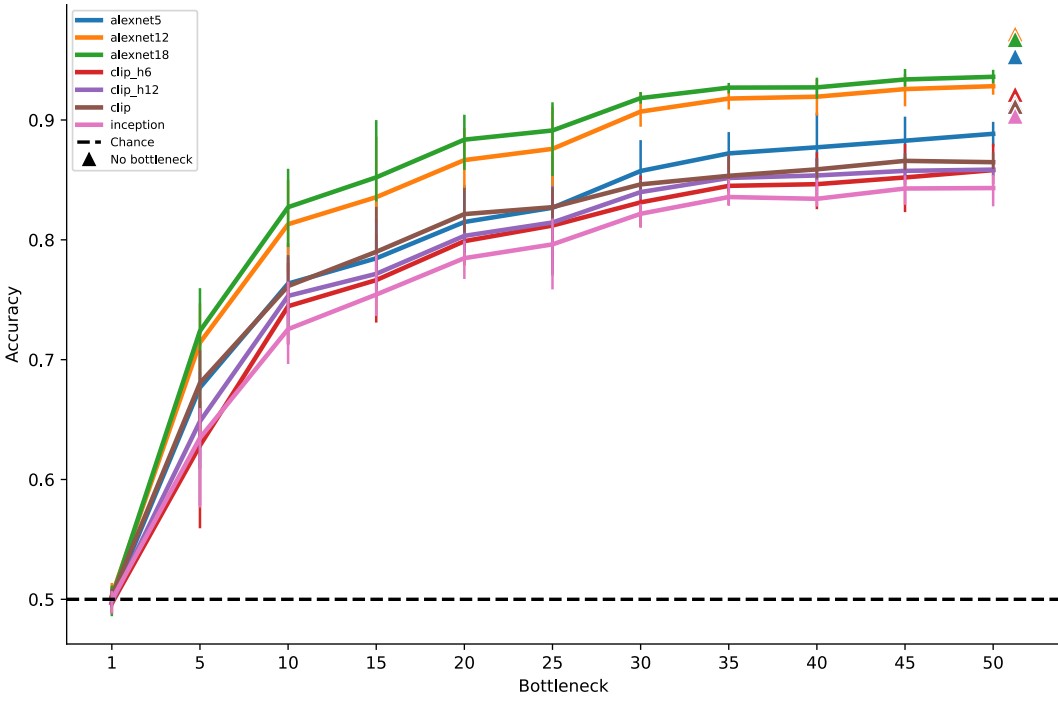

Figure 13: **BrainDiffuser identification accuracy.** We evaluate the agreement between latent embeddings of of ground-truth and decoded images of the BrainDiffuser method using the identification accuracy protocol described in [26].

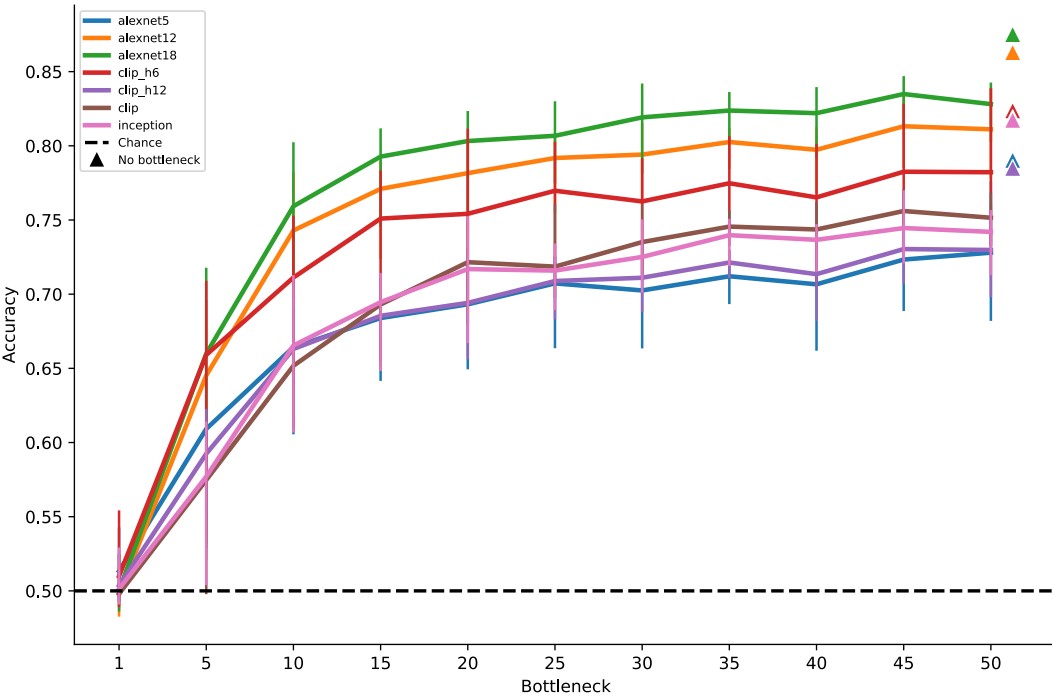

Figure 14: **Takagi identification accuracy.** We evaluate the agreement between latent embeddings of of ground-truth and decoded images of the Takagi method using the identification accuracy protocol described in [26].

## A.6   Takagi Image Grid with BrainDiffuser Image Indices

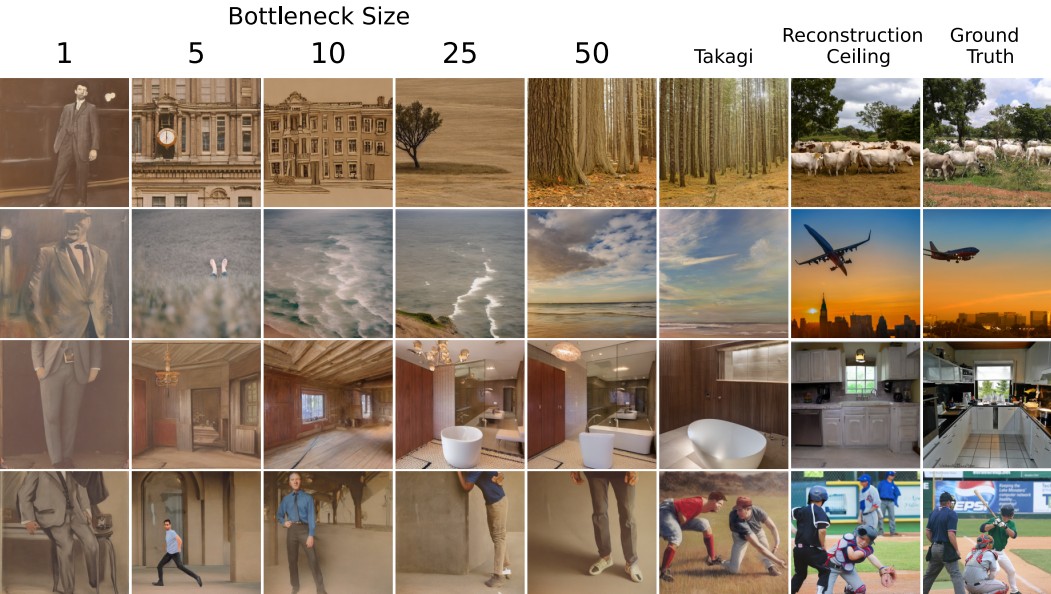

Figure 15: Generalization to other reconstruction methods: Shown here are images reconstructed for several bottleneck sizes applying our BrainBits approach to Takagi & Nishimoto et al.

## A.7 Effective dimensionality of the fMRI inputs

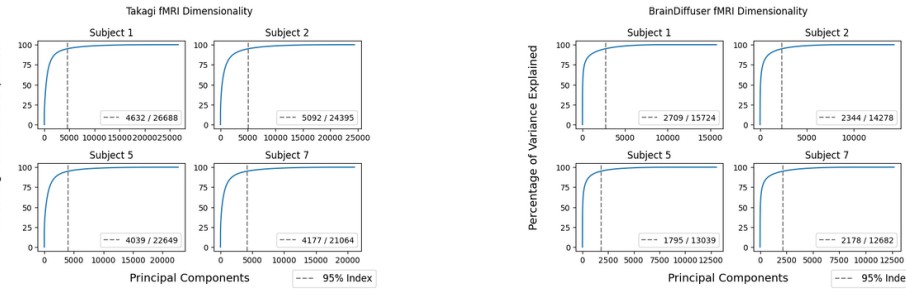

(a) brain-diffuser.      (b) Takagi et al.

Figure 16: Effective dimensionality (dashed line) of fMRI inputs as used in brain-diffuser (a) and Takagi et al. (b). Here, effective dimensionality is computed the same was as in the main text, as the number of dimensions needed to explain 95% of the variance (as determined by PCA).

