# OpenReview forum: "BrainBits: How Much of the Brain are Generative Reconstruction Methods Using?"
_NeurIPS.cc/2024/Conference — NeurIPS 2024 poster_

### Official Review · Reviewer_eVVq · 2024-07-05

**Soundness:** 2
**Presentation:** 2
**Contribution:** 2
**Rating:** 4
**Confidence:** 3

**Summary:**

This paper aims to reduce the dimensionality of brain signals using linear layers to determine the minimal dimension required to preserve most of the reconstruction quality. The study involves experiments with three existing methods: two for brain-to-image reconstruction and one for brain-to-language reconstruction.

**Strengths:**

- The paper is easy to follow.
- The paper aims to explore the extent of information necessary for brain decoding.

**Weaknesses:**

1. The paper suggests that better reconstruction may be achieved with less brain signal input, yet all brain signals are utilized by the reconstruction models. While compressing signals to lower dimensions is possible, it does not guarantee that fewer signals can be used as input.
2. Only diffusion-based models are employed for decoding images. The relatively favorable results may stem from well-trained diffusion models capable of operating with minimal semantics. The discrepancy in bottleneck dimensions between image and text decoding (50 vs. 1000) could support this. In reality, diffusion models exhibit significant randomness in image generation, enabling the selection of images resembling ground truth stimuli from various generation runs. Consequently, drawing general conclusions based on the use of diffusion-based models may be challenging.
3. The performance of brain-to-image generation remains unsatisfactory. While subject1 shows promising results in some instances, the images generated using whole brain signals lack the structure and detail seen in the stimulus images. Furthermore, in additional poor cases not illustrated in this paper, the decoding results may be even more unsatisfactory. Therefore, achieving good image decoding results with fewer signals may be challenging.
4. The number of samples for different subjects is limited. Literature suggests significant performance variations in image generation among subjects, with some individuals exhibiting poor reconstruction outcomes.
5. The paper asserts that the method can be adapted to various neural recording modalities; however, differences in temporal and spatial resolution as well as signal characteristics among recorded signals pose uncertainties about BrainBits' adaptability. Notably, no experiments support this claim.
6. The second paragraph in the introduction appears somewhat contradictory.  It mentions that higher quality reconstruction may require same or less signal from the brain, while also acknowledging the scarcity of open neuroscience datasets, especially those of sufficient scale to support this type of research.
7. The brain regions depicted in Figure 5 are not clearly labeled, making it difficult for readers to fully understand.
8. The effective dimensionality in Figure 4 lacks explicit clarification.
9. Text occlusion is apparent in both Figure 3 and Figure 11, affecting the clarity.

**Questions:**

1. Given that lines 4-7 of the abstract lack a clear connection with the following sentences, and the paper does not tackle issues related to understanding stimulus distribution and enhancing text or image reconstruction, what meaning does it actually convey?
2. Why were two fMRI plus diffusion models chosen for image decoding instead of experimenting with other signal types or generative models?
3. Do different subjects activate different voxels for the same stimulus image? Or do different voxels get activated for different stimulus images within a subject? If so, utilizing a fixed subset of voxel recordings as input becomes challenging due to the uncertainty of which voxels will be utilized later.
4. Can BrainBits operate with fewer brain signals as input? If not, what specific contributions does this paper offer to the community despite this limitation?

**Limitations:**

There is no potential negative societal impact.

---

> ### Author Rebuttal · Authors · 2024-08-07
>
> We thank the reviewer for their time and extensive feedback. We answer questions and clarify a few points below.
>
> - _The paper suggests that better reconstruction may be achieved with less brain signal input, yet all brain signals are utilized by the reconstruction models._
>   - Yes, while all brain signals are available at input, the dimension of the bottleneck puts a hard upper bound on the amount of information available at reconstruction time. Notably, increasing the size of this bottleneck leads to better performance, from which we infer that these bounds are tight.
>   - We have preliminary analysis of which parts of the brain are likely important (Figure 5), but truly selecting inputs by dropout is left for future work. This procedure is complicated by differences between subjects and significant computational costs, since dropout is a combinatorially difficult problem.
>
> - _Only diffusion-based models are employed for decoding images. The relatively favorable results may stem from well-trained diffusion models capable of operating with minimal semantics. Diffusion models exhibit significant randomness in image generation, enabling the selection of images resembling ground truth stimuli._
>   - We do not allow selecting images resembling the ground truth stimuli. We agree with the reviewer that doing so would be highly problematic. Instead, models get one chance to produce an image: one image for each of the 1000 stimuli. Then the average performance is computed. The variance on that average performance is small, since over so many images the stochastic nature of the diffusion models averages out.
>   - Regarding the power of diffusion models, we agree entirely! Our entire point is that these models operate with minimal semantics, rather than with extremely rich representations needed to actually encode the entire image. This is not a particular property of diffusion models, it is instead a property of any large model trained on data that is similar to the test data shown to subjects.
> - _The performance of brain-to-image generation remains unsatisfactory._
>   - I think we’re in agreement here! The main thrust of our argument is this: prior work has claimed that brain-to-image reconstruction is satisfactory to some degree, according to existing metrics.
>   - And whatever flaws these images have is up for discussion, but our experiments show that you can achieve the same performance on those metrics with much less information from the brain. This is so little information that clearly something must be missing from the images. Yet this point has not been made in the literature and it is critical for further progress.
> - _The number of samples for different subjects is limited._
>   - Our paper should be thought of as a meta-study over the field of work in brain-to-image reconstruction. To this end, we consider those subjects and samples used by existing works.
> - _Differences in temporal and spatial resolution as well as signal characteristics among recorded signals pose uncertainties about BrainBits' adaptability._
>   - Our method is very general: and simply entails adding a restriction on information flow through the reconstruction pipeline. This is applicable, no matter the modality.
> - _Authors claim reconstruction may require same or less signal from the brain, while also acknowledging the scarcity of open neuroscience datasets._
>   - To clarify, there is a distinction between the signal available in a single brain reading and the data available in the training dataset. The first is relevant to the quality of reconstruction, the second is relevant to the ease of training a reconstruction method.
>
> - _Regions depicted in Figure 5 are not clearly labeled_
>   - We created a legend with labeled regions (see additional PDF) that we will add to the main text.
> - _The effective dimensionality in Figure 4 lacks explicit clarification._
>   - Lines 201 through 204 explain the effective dimensionality method used. We will update the caption to include this.
> - _Text occlusion in both Figure 3 and Figure 11._
>   - Thanks! We will add labels, clarification in the caption for Figure 4, and we will fix the spacing in Figures 3 and 11
> - _What meaning do lines 4-7 convey?_
>   - The reasoning is this: currently only measuring the quality of the reconstructed images allows for models to improve for reasons that do not involve extracting more signal for the brain. Lines 4-7 simply list a few of these possible reasons. The purpose of our work is to introduce a metric that cannot be “gamed” in such ways.
> - _Experimenting with other signal types or generative models?_
>   - The overwhelming number of recent state of the art publications in image reconstruction use fMRI enabled by the NSD dataset. Since this is by far the most common paradigm, we decided to adopt it. There is nothing about our method that is signal-specific. Nor is there anything that is specific to diffusion generative models. Given the state of the field, we chose the most representative methods.
> - _Do different voxels get activated for different stimulus images within a subject?_
>   - All of our learned maps and results are per subject. This avoids this problem.
> - _Can BrainBits operate with fewer brain signals as input? If not, what specific contributions does this paper offer to the community?_
>   - We are not sure if we understand the question. BrainBits is not a reconstruction method in itself. BrainBits is a method for measuring the dimensionality of the neural recordings required to perform reconstruction. And it is applicable, no matter the input size. It addresses a key problem: large networks can reconstruct images well with little information by exploiting the similarity between their massive training sets and relatively restricted test sets. Without BrainBits, one cannot disentangle why a model is performing better. Is it explaining more of the brain or taking more advantage of its priors? We show that these priors can be extreme.

---

> > ### Comment · Reviewer_eVVq · 2024-08-11
> >
> > Thank you for your responses.
> >
> > As analysis of which parts of the brain have been provided in Figure 5, why not generate images using the brain activities from these brain regions? This could strongly support the main idea of this paper, while the other evidences seems to be circumstantial.

---

> > > ### Author Response · Authors · 2024-08-13
> > > **Response to further comments**
> > >
> > > This is a really good idea for future work, and precisely what we are considering for our sequel! But the evidence as it stands is not at all circumstantial. The information bottleneck puts a hard upper-bound on the amount of signal that can be used for reconstruction. And the fact that increasing the bottleneck leads to increased image similarity shows that this bound is tight for sizes 1-50. Of course, there are many ways that the contents of this bottleneck can be further characterized, as you point out, and this is a good direction for future work.

---

> > > > ### Comment · Reviewer_eVVq · 2024-08-13
> > > >
> > > > Thanks for the response. I have no further concerns. While I have adjusted my score, I haven't raised it to a higher level as I am still uncertain about the work's overall significance in the brain decoding field. As I mentioned above, validating the successful generation of images based on brain activity from specific brain regions would be more beneficial.

---

### Official Review · Reviewer_eceS · 2024-07-06

**Soundness:** 4
**Presentation:** 4
**Contribution:** 3
**Rating:** 7
**Confidence:** 4

**Summary:**

The authors introduce BrainBits, an information-bottleneck pipeline that measures reconstruction performances from brain signals (fMRI datasets) as a function of bottleneck size by (linearly) projecting the data into a lower dimensional space of controlled dimensionality. The rationale is to disentangle the contributions to improved reconstruction quality seen in recent  works into (1) improvements in actual decoding (better use of neural information - the actual goal of the decoding techniques) and (2) general improvements in generative models (more powerful architectures with better priors). Indeed the authors reveal that modern improvements are cause by the latter, with recent models making very little use of neural signals. Furthermore, the BrainBits pipeline enables inspection of which brain areas are mostly relied upon. The technique has broad applicability to both vision and language decoding and different signal modalities (however, only fMRI is reported).

**Strengths:**

*a. Originality:* The work provides a novel method to rigorously characterize decoding performances which surpasses previous metrics on critical aspect and is resilient to simply scaling decoder complexity.
*b. Quality:* The work quality is generally high, with enough experimental result to support the author's claim.
*c. Clarity:* The work is well written and properly organized.
*d. Significance:* The results presented are of high significance for the brain-decoding community. Recent years have witnessed a wealth of novel contributions, each time showcasing evermore detailed brain reconstructions, implying significant advancements in our ability to decode neural signals. This work offers a much needed warning that we need to guard against misleading metric improvements granted by more powerful generative models and provides the tools to do so. The proposed pipeline is flexible and easy to build upon, offering a valuable contribution to the community at large.

**Weaknesses:**

- Figure 3c is hardly readable. Why are the axes y-scale range in [0, 1] (which then requires smaller inset to actually inspect the data)? Can't the range by set according to data dynamic range?
- Regarding the analysis on brain regions (Figure 5). The authors claim that the BrainDiffuser model "*As the bottleneck size goes up models exploit those original areas but do not meaningfully expand to new areas*". By looking at the (small) image, however, bottleneck size 50 seems to have far fewer "silent voxels" (dark purple) than bottleneck 1 or 5 for example. This appears to hold also for other examples presented in Figure 11 in the Appendix (Note that Figure 11 has cut-out subplot titles that are illegible). Why do authors claim that the measured expansion is not "meaningful"? What would a meaningful expansion look like?

**Questions:**

- What are the implications of neural coding redundancy on the analysis presented in Fig. 4 where it is shown that a small fraction of the bottleneck dimension are effectively used by the models? Could it be that indeed the underlying neural code is highly redundant hence the model can make effective use of neural signals by using a much lower dimensionality?
- Authors propose that each method should report a reconstruction ceiling. However it appears that such ceiling is not always clear how to measure. For example authors say: "*No analogous ceiling procedure exists for the language reconstruction method, Tang et al 2023, [...]*".  It is my understanding that authors do not offer a general procedure to deal with this problem. If that's correct, how would they suggest to approach this problem in general?

**Limitations:**

The authors have adequately addressed the limitations of their study by presenting a dedicated section (6) with extensive discussion.

---

> ### Author Rebuttal · Authors · 2024-08-07
>
> We appreciate the reviewer's time and good questions, which helped us clarify our interpretations. We're glad that the reviewer found our paper important to the field and easy to follow. Below, we answer the posted questions.
>
> - _Can't the range be set according to data dynamic range?_
>   - The inset is to emphasize the fact that for the text decoding case, reconstructed text is still very close to chance in an absolute sense. We will enlarge the label text for legibility and include a larger plot in the supplement.
> - _Why do authors claim that the measured expansion is not "meaningful"? What would a meaningful expansion look like?_
>   - In the 5 vs 50 case we meant to point out that no new significant cluster areas of activity emerge. You are correct in that “meaningfully expand” is not very precise, we will update the language in our paper to better describe our observations. We merely meant to observe that the most salient regions remained the brightest across all scales, and few new islands of activity were highlighted for the larger bottlenecks.
> - _Figure 11 has cut-out subplot titles that are illegible_
>   - Thanks for the catch! We’ll fix the spacing. To clarify, the rows of that figure were in subject ID order: 1, 2, 5, 7.
> - _What are the implications of neural coding redundancy on the analysis presented in Fig. 4 where it is shown that a small fraction of the bottleneck dimension are effectively used by the models? Could it be that indeed the underlying neural code is highly redundant hence the model can make effective use of neural signals by using a much lower dimensionality?_
>   - For comparison, for brain-diffuser, the average effective dim of the fMRI inputs is 2,257. For Takagi et al, the average is 4,485 (see additional PDF). This suggests that the underlying neural code is far higher dimensional than our bottleneck size, despite any redundancies that may exist.
> - _Authors propose that each method should report a reconstruction ceiling. How would they suggest to approach this problem in general?_
>   - The very simplest version of incorporating a ceiling, would be to always show the evaluation metrics, as they are with the ground truth inputs. For example, we show the complete scale from 0.0 to 1.0 on Figure 3c. We argue that looking at the distance to this simple ceiling is an important part of gauging whether reconstruction results can be called successful or not. A slightly more sophisticated ceiling involves using the ground truth image latents as conditioning information for the diffusion model. This is what we do in Figure 3a-b. Both of these procedures should always be feasible.

---

### Official Review · Reviewer_khsL · 2024-07-08

**Soundness:** 2
**Presentation:** 3
**Contribution:** 3
**Rating:** 6
**Confidence:** 4

**Summary:**

The paper proposes a method called BrainBits which aims at answering if the progress of the fMRI-to-Image/Text field of research comes from a better signal extraction from the brain or from other sources such as having better generative models or exploiting bad metrics. Their method introduces a bottleneck between the fmri data and various fMRI-to-Image/Text methods. Their method is thus directly applicable to basically every reconstruction method, and allows to change the dimensionality of the bottleneck and see whether one can obtain a substantial percentage of the original performance even with the bottleneck.

**Strengths:**

- The questions asked by the authors is very important to the field. Much progress has been made recently, but the causes of that progress remain unclear. Everyone would love the cause to be better brain signal extraction, and thus better understanding of the brain activity by the model. By providing a way to establish whether it is the case or not, the authors tackle a crucial issue. It is a known fact that the metrics used in the field lack a way of knowing how the brain data has been exploited.

- The results obtained by the authors are surprising: in most cases, even with a narrow bottleneck, we can obtain a substantial percentage of the performance of the original model.

- The paper is well written and easy to follow. The experiments are clear.

**Weaknesses:**

- The main weakness is that the bottleneck introduced in the paper does not definitely answer how much of the brain signal has been exploited in the process. Indeed, the MLP projecting the fMRI data to the bottleneck actually learns to identify the most important features within the brain data, in order to obtain the best reconstruction. Much like a VAE, it learns to compress information as efficiently as possible in order to have the best performance possible. Thus, the compression ratio, even if it is generally very high in the authors' results, is also a result of the best effort of the MLP to identify the best features. This crucial point kind of defeats the point of the paper: isolating how much of the brain signal was extracted and used. However, given the size of the compression ratio (300 in the case of BrainDiffusers for instance), I still believe that the authors rightfully identified that not all of the brain signal is used, and that their research is on the right track. However this point could be further improved and disentangled.

**Questions:**

Please answer to the main weakness I have identified. Any convincing clarification would result in a improved score of my review.

**Limitations:**

I would add the limitation identified in the weakness part, if it still stands after the rebuttal.

---

> ### Author Rebuttal · Authors · 2024-08-07
>
> We thank the reviewer for recognizing the importance of our work and for their time and feedback.
>
> - _The main weakness is that the bottleneck introduced in the paper does not definitely answer how much of the brain signal has been exploited in the process. Indeed, the MLP projecting the fMRI data to the bottleneck actually learns to identify the most important features within the brain data, in order to obtain the best reconstruction. Much like a VAE, it learns to compress information as efficiently as possible in order to have the best performance possible. Thus, the compression ratio, even if it is generally very high in the authors' results, is also a result of the best effort of the MLP to identify the best features. This crucial point kind of defeats the point of the paper: isolating how much of the brain signal was extracted and used. However, given the size of the compression ratio (300 in the case of BrainDiffusers for instance), I still believe that the authors rightfully identified that not all of the brain signal is used, and that their research is on the right track. However this point could be further improved and disentangled._
>
>   - First, to clarify, we use a single linear layer, not an MLP, for projection. The nice thing about using a single linear mapping is that we can be fairly confident that no extra expressive power has been added to the reconstruction method. A single linear layer has much less representational power than a VAE. Of course, it is likely that better compressions may be possible with more powerful projection mappings, but these would be less interpretable as the reviewer correctly points out.

---

> ### Author Response · Authors · 2024-08-13
> **Let us know if we can answer anything**
>
> We greatly appreciate your time in reviewing our paper. Please let us know if our response helped clear anything up, or if there is anything else you would like answered!

---

### Official Review · Reviewer_BfHp · 2024-07-26

**Soundness:** 3
**Presentation:** 2
**Contribution:** 3
**Rating:** 6
**Confidence:** 4

**Summary:**

The paper presents a method called BrainBits that aims to assess the extent to which generative image reconstruction based on fMRI data is based on the neural data itself, versus some spurious contribution of the reconstruction model itself (e.g., a stronger prior over natural images, or overfitting to the distribution of images used in benchmarks). The method involves introducing a bottleneck of varying size, and assessing how reconstruction performances varies based on the size of the bottleneck. They apply their method to two image reconstruction tasks and one language reconstruction task. They find that performance plateaus at a surprisingly small bottleneck size, and use this finding to argue that neural stimulus reconstruction approaches are only using a fraction of the information available in the neural data, such that we should worry that recent improvements in reconstruction are due to contributions from models, rather than more effective extraction of information from the brain.

**Strengths:**

The motivation of the paper is timely, sound, and convincing: we need to interrogate the possible sources of improved stimulus reconstruction performance, and we need ways to assess the contribution of the model versus the neural data itself. The authors argue this point clearly, and I could imagine this paper playing a useful role in raising awareness of this problem in the field, and making a first stab at addressing this issue.

On a methods level, the paper appears to be sound: including the random performance and reconstruction ceiling provides helpful context, and the analyses linking the bottleneck activations to brain topography and decodable features were illuminating. The figures were well-chosen and clearly presented. The analyses support the case that reconstruction performance plateaus at a small number of dimensions (although I think the authors draw inferences from this that aren't warranted; see weaknesses section).

**Weaknesses:**

While I found the motivation for the paper to be convincing, and I believe the authors effectively make the case that a relatively small number of dimensions is sufficient to achieve maximum reconstruction performance, I think the authors' argument ignores an important piece of the puzzle: what is the effective dimensionality of the actual neural activations? For the sake of argument, suppose that the dimensionality of the neural responses (over the space of stimuli sampled) is 50: then, the fact that reconstruction plateaus with a bottleneck of size 50 is not due to the method using a small proportion of the underlying information available in the neural signal, but rather due to the fact that the neural activity patterns are themselves low dimensional (e.g., due to correlations in the underlying neuronal firing patterns, or the fact that the BOLD signal in each voxel reflects the aggregate activity of many neurons). Without addressing this issue, I don't think the authors' conclusions follow from their results.

The writing style sometimes borders on overly informal ("Although, small bottlenecks are perhaps not that
interesting given that the goal is to explain more of the brain"), though I believe this is easily fixed with further edits.

**Questions:**

1. How can we know that the plateau in performance at a small bottleneck size reflects the model ignoring usable information in the neural measurements, versus exhausting the usable information available in the neural measurements? As mentioned in the weaknesses section, this is hard to assess without measuring the dimensionality of the neural activations.

2. Under the hypothesis that the prior in generative models is strongly contributing to reconstruction performance, it is odd to me that higher-level visual areas don't seem to be used by the reconstruction pipeline, even at higher bottleneck sizes: given that their activation is highly informative regarding object category, and given that these models presumably have a strong prior for how objects tend to look, I wouldn't have predicted this. It would be interesting (though perhaps too much for the scope of this paper) to restrict the pipeline to particular sectors of the visual system, and examine how this affects performance.

**Limitations:**

There are no negative societal impacts that I can think of, and the limitations section provides helpful context regarding the practical application of their method.

---

> ### Author Rebuttal · Authors · 2024-08-07
>
> Thank you for your time and feedback. We're glad you found our work timely, sound, and convincing. We appreciate your questions, which have helped us sharpen our descriptions.
>
> - _What is the effective dimensionality of the actual neural activations? How can we know that the plateau in performance at a small bottleneck size reflects the model ignoring usable information in the neural measurements, versus exhausting the usable information available in the neural measurements?_
>   - The reviewer asks a good question! What is the dimensionality of the input fMRI signal and are we simply recovering this number? For brain-diffuser, the average effective dim of the inputs is 2,257. For Takagi et al, the average is 4,485. In comparison, 50 is small. (See additional PDF). We will add this figure to the appendix and note this computation in the main text.
>   - We are glad the reviewer found our paper well motivated. We would like to further add that our main contention is not simply that the amount of needed brain data is small in an absolute sense, but that, as a field, we should always use a metric that is sensitive to the amount of brain data used for reconstruction, so as not to be misled by methods that improve on the image prior without improving on the extraction of neural signal.
>
> - _Under the hypothesis that the prior in generative models is strongly contributing to reconstruction performance, it is odd to me that higher-level visual areas don't seem to be used by the reconstruction pipeline, even at higher bottleneck sizes: given that their activation is highly informative regarding object category, and given that these models presumably have a strong prior for how objects tend to look, I wouldn't have predicted this. It would be interesting (though perhaps too much for the scope of this paper) to restrict the pipeline to particular sectors of the visual system, and examine how this affects performance._
>   - We suspect that higher level areas aren’t being used in part because they are redundant with early areas. Representations in early areas are retinotopic and potentially simpler than those in later areas which may be neither.
>   - And yes! This is exactly what we aim to do in subsequent work and the main other application of our method: probing the information available in the decodings conditioned on different areas of the brain. We plan to automate this by having models look at the resulting images and then quantify the information that is being decoded: like albedo, lighting, shape, texture, class, relationships, etc.

---

> > ### Comment · Reviewer_BfHp · 2024-08-07
> >
> > Thank you to the authors for response, which has clarified some of my questions with the submission:
> >
> > *"The reviewer asks a good question! What is the dimensionality of the input fMRI signal and are we simply recovering this number? For brain-diffuser, the average effective dim of the inputs is 2,257. For Takagi et al, the average is 4,485. In comparison, 50 is small. (See additional PDF). We will add this figure to the appendix and note this computation in the main text."*
> >
> > Thank you for these further analyses. However, a bit more clarity in how they're described would be useful. The PDF says "dimensionality of the fMRI inputs"--is this referring to the beta values, or to the raw time series values? And, to what extent do these dimensions reflect stimulus-related variability, versus the dimensionality of random sources of noise? This question is important, since only the former is useful for stimulus reconstructions, and only this is informative regarding how much usable information is being used by the networks. If many of these dimensions are noise-related, it is unsurprising that they're being disregarded by these reconstruction pipelines. A bit more clarity on this point would resolve my concerns on this issue.
> >
> > *"We suspect that higher level areas aren’t being used in part because they are redundant with early areas. Representations in early areas are retinotopic and potentially simpler than those in later areas which may be neither."*
> >
> > I remain puzzled by this issue, and I think it would be interesting to further explore (BOLD is low-res, and I would've expected the high-level information to help disambiguate parts of the image that are fuzzy based on early visual regions), but I think the paper is fine without definitively answering this question.

---

> > > ### Author Response · Authors · 2024-08-09
> > > **Response to follow up comment**
> > >
> > > - _To what extent do these dimensions reflect stimulus-related variability, versus the dimensionality of random sources of noise?_
> > >   - This is an open question for the field at large. Parsing out exactly which activations are pertinent only to the task, is a yet unanswerable question. But this is not at odds with the goal of BrainBits! On the contrary, the purpose of BrainBits as a metric is to identify when reconstruction has ceased to extract signal from the neural recordings, whatever dimensionality that signal may be, and begun to make improvements purely on the image generation prior. The fact that this particular question remains open makes the need for BrainBits all the more critical because otherwise the only way to compute stimulus-specific ID is to attempt to find the ID of the corresponding neural response. As you note, this will possibly contain activity unrelated to the stimulus. BrainBits offers a much better alternative: try to find the ID of the bottleneck necessary for reconstructing the stimulus.
> > > - _Is this referring to the beta values, or to the raw time series values?_
> > >   - For the image reconstruction methods we computed the effective dimension of the betas, specifically the “betas_fithrf_GLMdenoise_RR (beta version 3; b3)” provided by the NSD dataset (masked to visual areas as was done by the reconstruction methods we investigated). These betas were computed using the GLM single method described in “Jacob S Prince, Ian Charest, Jan W Kurzawski, John A Pyles, Michael J Tarr, Kendrick N Kay (2022) Improving the accuracy of single-trial fMRI response estimates using GLMsingle eLife 11:e77599” which fits per voxel heart rate functions and attempts to remove other noise using the multiple responses available per stimuli. We also averaged the betas across repeated presentations within the subject. This is the same data and procedure that was used by the reconstruction methods we investigated.
> > >   - From the NSD data manual the betas are described as: “betas_fithrf_GLMdenoise_RR (beta version 3; b3) – GLM in which the HRF is estimated for each voxel, the GLMdenoise technique is used for denoising, and ridge regression is used to better estimate the single-trial betas.”
> > >
> > > - _I remain puzzled by this issue, and I think it would be interesting to further explore (BOLD is low-res, and I would've expected the high-level information to help disambiguate parts of the image that are fuzzy based on early visual regions), but I think the paper is fine without definitively answering this question._
> > >   - This is certainly an interesting question to consider! And the fact that it’s on the table is a strength of the general approach. We plan on looking into it, along with looking at different regions, in a followup publication.

---

> > > > ### Comment · Reviewer_BfHp · 2024-08-12
> > > >
> > > > Thank you for the further clarifications. I think as long as you clarify the methods in the way I mentioned, my issues with the paper are now addressed.

---

### Author Rebuttal · Authors · 2024-08-07

We thank the reviewers for their time and helpful feedback! We address each reviewer's concerns individually.

---

### Decision · Program_Chairs · 2024-09-25

**Decision:**

Accept (poster)

**Comment:**

This study introduces a low-dimensional bottleneck on the information utilized for seen-image reconstruction from fMRI responses. The study demonstrates that even low-dimensional bottlenecks can yield seemingly good reconstruction performance, suggesting that much of the success of current methods may rely more on strong priors of the image-generating models than on detailed neural information readout. Most reviewers and the AC agreed that this work is important, as it is the first to tackle a major problem plaguing the subfield of seen-image reconstruction.

Questions remain about the approach taken in this work to limit the read-out information. In principle, a low-dimensional bottleneck does not constrain the amount of information utilized; while achieving good reconstruction performance with a compressed, low-dimensional readout is remarkable, it does not prove that only a specific amount of information was used. As suggested in the paper's discussion, a quantized bottleneck would have provided stronger control over the information used by restricting it to a defined number of bits of Shannon information, aligning more closely with the paper's title. Additionally, one reviewer expressed concerns about the quality of the reconstructions and the general applicability of the approach.

Given the importance of disentangling the contributions of image priors and actual neural decoding, most reviewers and the AC agree that this work should be accepted to NeurIPS, despite its limitations. We anticipate a fruitful debate and follow-up studies that will refine the methodology proposed in this work.